# Detection of Drowsiness and Impending Microsleep from Eye Movements

**Silvia Makowski**\*                                                    SILVIA.MAKOWSKI@UNI-POTSDAM.DE
**Paul Prasse**\*                                                          PAUL.PRASSE@UNI-POTSDAM.DE
*University of Potsdam, Potsdam, Germany*

**Lena Ann Jäger**                                                          JAEGER@CL.UZH.CH
*University of Zurich, Zurich, Switzerland*
*University of Potsdam, Potsdam, Germany*

**Tobias Scheffer**                                               TOBIAS.SCHEFFER@UNI-POTSDAM.DE
*University of Potsdam, Potsdam, Germany*

**Editor:** Editor's name

## Abstract

Drowsiness is a contributing factor in an estimated 12% of all road traffic fatalities. It is known that drowsiness directly affects oculomotor control. We therefore investigate whether drowsiness can be detected based on eye movements. To this end, we develop deep neural sequence models that exploit a person's raw eye-gaze and eye-closure signals to detect drowsiness. We explore three measures of drowsiness ground truth: a widely-used sleepiness self-assessment, reaction time, and impending microsleep in the near future. We find that our sequence models are able to detect drowsiness and outperform a baseline processing established engineered features. We also find that the risk of a microsleep event in the near future can be predicted more accurately than the sleepiness self-assessment or the reaction time. Moreover, a model that has been trained on predicting microsleep also excels at predicting self-assessed sleepiness in a cross-task evaluation, which indicates that upcoming microsleep is a less noisy proxy of the drowsiness ground truth. We investigate the relative contribution of eye-closure and gaze information to the model's performance. In order to make the topic of drowsiness detection more accessible to the research community, we collect and share eye-gaze data with participants in baseline and sleep-deprived states.

## 1. Introduction

The term *drowsiness* refers to the transition from being clearly awake to being clearly asleep. This transition phase is already characterized by profound changes in motor control, cognition, brain activity, and consciousness (McGinley et al., 2015). The European Commission estimates that drowsiness is a contributing factor in 12% of all road accidents (Directorate-General for Mobility and Transport of the European Union, 2023)—these accidents account for 162,000 of the 1.35 million annual road traffic fatalities (Center of Desease Control, 2023).

Driver cameras that are entering the automotive market are capable of extracting the driver's eye closure and eye gaze (Halin et al., 2021). It is known that fatigue and drowsiness are associated with an increased blink frequency (Santamaria and Chiappa, 1987; Luckiesh and Moss, 1937; Hoffman, 1946); this connection may be understood as cessation of the

---

\* Both authors contributed equally to this research.

attention-driven suppression of blinks (Schleicher et al., 2008). Blink duration (Morris and Miller, 1996; Häkkänen et al., 1999) and the individual standard deviation of blink rates and duration increase with increasing fatigue (Schleicher et al., 2008). Motivated by these psychological findings, known approaches to drowsiness detection use features that are derived from the eye-lid closure—such as blink frequency, duration, and eye-lid velocity (Wilkinson et al., 2013; Horng et al., 2004; Nguyen et al., 2015; Rumagit et al., 2017). The percentage of time in which the pupil is covered by the lid (PERCLOS) (Skipper and Wierwille, 1986) tends to increase with increasing fatigue; however, individuals can willingly keep their eyes open despite being fatigued, and even while exhibiting indicators of sleep in the EEG (O'Hanlon and Kelley, 1977).

Psychological research has found features of gaze events, such as saccades and fixational micro-movements to be correlated to drowsiness. The saccadic accuracy and peak saccadic velocity can be negatively impacted by fatigue, but with a high inter-subject variability (Galley, 1989; Hirvonen et al., 2010). Di Stasi et al. (2013) find that ocular stability decreases as a function of mental fatigue: In a visual search task, the velocity of saccades and micro-saccades decreases under fatigue whereas the velocity of ocular drift increases. In a driving-simulator experiment, saccadic duration increases, saccadic speed decreases, and their standard deviations increase with increasing fatigue (Schleicher et al., 2008). These psychological findings, and the fact that the eye gaze can be measured by optical sensors, motivate us to explore eye-movements in addition to eye-closure signals as a predictor of drowsiness.

However, when developing machine learning methods for drowsiness detection, obtaining valid ground-truth labels poses a major methodological challenge. Whereas the state in which a person is clearly sleeping (sleep stage 2) can be unambiguously labeled in an electroencephalogram (EEG) recording that shows the characteristic sleep spindles or K complexes, drowsiness (sleep stage 1) cannot be directly observed. Although drowsiness has long been known to be associated with certain changes in the EEG signal (Matousek and Petersén, 1983; Dement and Kleitman, 1957; Santamaria and Chiappa, 1987), it cannot be unambiguously detected from the EEG signal alone across different individuals, but additional motion or eye movement data is necessary (Moser et al., 2009; Santamaria and Chiappa, 1987), which not only makes the manual labeling difficult (Rechtschaffen and A, 1968; Berry et al., 2020), but has led to an overwhelming number of features extracted from the EEG signal that, at the group level, all have been shown to correlate with drowsiness (Stancin et al., 2021). Moreover, in a task such as driving, the EEG signal is heavily contaminated by muscle artifacts. In sum, although for the detection of sleep stage 2 and higher, EEG is the gold-standard measurement technique to determine ground-truth values, it is not an ideal tool to determine drowsiness ground truth.

The *Karolinska sleepiness scale (KSS)* (Åkerstedt and Gillberg, 1990) is a widely-used self-assessment, in which participants are asked in regular intervals to rate their fatigue on a scale from "1—very alert" to "9—very sleepy, great effort to keep alert, fighting sleep". The KSS self-assessment is generally considered as a gold-standard proxy of drowsiness ground truth. It is correlated to driving errors and EEG-derived indicators of fatigue (Kaida et al., 2006). The drawback of the KSS score is that it is strongly subjective, and that participants can easily misjudge their own level of drowsiness.

The *psychomotor vigilance test (PVT)* (Dinges and Powell, 1985) measures users' reaction time during a ten-minute repetitive reaction test. While the PVT offers an objective, quantitative measure of vigilance, it interrupts any other user activity for ten minutes, and the functional relationship between vigilance and reaction time is highly individual.

This paper makes a number of contributions.

1. We introduce upcoming microsleep events in the near future as a new proxy of drowsiness ground-truth. We argue that upcoming microsleep is a highly relevant proxy of the drowsiness ground truth for applications such as driver monitoring, because prolonged driving without visual perception is objectively hazardous.

2. We show that the risk of impending microsleep can be predicted more accurately than both KSS self-assessment and reaction time. Moreover, a model trained to predict microsleep is at least as good at predicting KSS assessments as a model trained on KSS scores, which indicates that upcoming microsleep is a less noisy form of ground truth.

3. We develop CNN, LSTM, and Bi-LSTM neural network architectures that directly process the raw eye-closure and eye-gaze signal to predict drowsiness by learning to extract the relevant information from the input signal in a fully data-driven way. As reference baseline, we implement an exhaustive list of published hand-crafted features that serves as input to a random-forest classifier.

4. In an ablation study, we quantify the relative contributions of eye-closure and eye-gaze features to fatigue detection.

5. In order to make the topic of drowsiness detection more accessible to the research community, we collect and share a database of 47 participants in baseline and sleep-deprived states with KSS, PVT, and impending microsleep ground truth. We also implement an exhaustive collection of published engineered eye-closure and eye-gaze features and share their implementation.

The remainder of this paper is structured as follows. Section 2 lays out the problem setting, Section 3 introduces the drowsiness detection models, Section 4 reports on our data collection. In Section 5 we present the experimental results. Section 6 discusses the results and related work. Section 7 concludes.

## 2. Problem Setting

In all the variations of problem settings that we study, the input to the system consists of the following signals:

- A sequence of raw eye gaze yaw and pitch angles of the left and right eye over the observation period, recorded by a video-based eye tracker;

- an eye-closure signal on a scale of zero to one, where zero indicates an aperture of 12 mm or more, and one indicates fully closed eyes;

- an eye-state variable that indicates whether the pupil is covered by the eye-lid with values "open" (pupil not covered), "closed" (pupil covered), "partially closed", "not visible" (covered by an occlusion), "downcast" (head is pitched downward and either may either be closed or looking downward), and "not available" (tracking failure);

- an eye movement events signal that indicates the presence of a fixation or saccade. Fixations are phases of relative stability during which only micro-movements occur and visual information is perceived, whereas saccades are fast relocation movements during which information uptake is suppressed.

While drowsiness ground truth cannot be observed directly, we will study a new proxy, as well as reference proxies. The first and most common reference proxy for drowsiness ground truth is the KSS self-assessment score (Åkerstedt and Gillberg, 1990) on a scale from "1–extremely alert" to "9–very sleepy, great effort to keep alert, fighting sleep". We study KSS prediction as a binary classification problem where the positive class is the aggregate of scores 7 through 9—the *sleepy* range—and the negative class is scores 1 through 6—the *alert* range. Due to the subjective nature of the KSS scale, it suffers from inter-subject variance that constitutes a principal upper bound on the accuracy that any system can possibly achieve.

As the second reference proxy of drowsiness ground truth, we consider the task of predicting the reaction time of the participants during the PVT task. In the PVT task, a small red dot appears in the center of a black screen after random time intervals. Participants have to press a button as soon as they recognize the dot. The reaction time is measured as the average interval between appearance of the dot and activation of the button. A person's reaction time is highly individual and, as Section 4.2 will confirm, some individuals can react faster while fighting sleep than others in their fully alert state.

We will therefore investigate the novel task of predicting impending microsleep events in a time window of the next 10 seconds. Definitions of microsleep events in the literature vary; we use a typical definition which is a continuous eye closure of at least 1,000 ms duration. *Predicting* microsleep events in the future must not be confused with the easier task of *detecting* microsleep. By the time an ongoing microsleep event can be *detected*, a hazardous situation is already in progress.

We will evaluate this task in two levels of difficulty. In addition to the evaluation of *all cases*, we will separately evaluate *hard cases*. The latter evaluation is restricted to positives in which the observation window does *not yet* contain a microsleep episode. Hard cases are first occurrences of microsleep that have to be predicted without the benefit of having observed preceding microsleep events that already provide evidence of drowsiness. Prediction of impending microsleep episodes is arguably linked closer to applications such as driver monitoring than estimating the KSS self-assessment score, because prolonged driving without visual perception is objectively hazardous.

For all binary output signals, we measure false-positive and true-positive rates. Each time step of each evaluation sequence constitutes an instance; in our evaluation protocol, time steps progress with a stride of 5 seconds. Depending on the target variable, a positive instance is a time step in which the model estimates the KSS as 7 to 9, or predicts an impending microsleep event, respectively. If the output matches the ground truth, the instance counts as a true positive, otherwise it is a false positive. All models under inves-

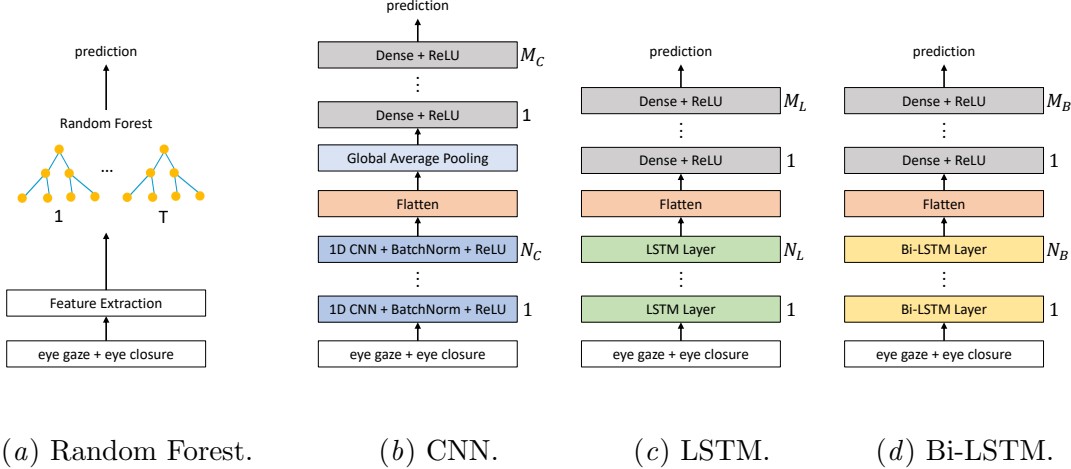

$(a)$ Random Forest.    $(b)$ CNN.    $(c)$ LSTM.    $(d)$ Bi-LSTM.

Figure 1: Drowsiness detection models.

tigation implement a continuous decision function; a positive output is triggered when the decision function exceeds a threshold. Adjusting this threshold changes the trade-off between false-positive and false-negative rates. The attainable pairs of true and false positive rates can be visualized in a ROC curve and aggregated in AUC values. The PVT reaction time is a continuous signal that we model as a regression task. We evaluate this task in terms of the *Root Mean Squared Error (RMSE)* and the *Coefficient of Determination $R^2$*.

## 3. Drowsiness Detection Models

This section introduces the models used to predict the impending microsleep events and the reference ground-truth proxies of the KSS self assessment and reaction time (see Figure 1).

### 3.1. Neural Networks

This subsection presents the proposed neural networks for eye-gaze based drowsiness detection. All presented neural networks take an input sequence of 60 seconds at 200 Hz, which results in 12,000 time steps. The input signals—described in detail in Section 2— result in 13 input channels: one eye-closure channel, seven channels that encode a discrete eye state, eye-gaze yaw and pitch velocities, and indicator channels for fixations, saccades, and missing values. The eye-state and eye-closure signals are generated by a commercial driver-monitoring system.

#### 3.1.1. CNN

We develop a one-dimensional CNN model architecture consisting of multiple CNN layers followed by a global average pooling layer and several fully connected layers that is designed to extract both local and global patterns in the eye-closure and eye-gaze signal (see Figure 1$(b)$).

The first layers of the model are one-dimensional convolutional layers, which are designed to learn local patterns in the data, followed by a batch normalization and the ReLU activation function. After the convolutional layers, we apply a global average pooling layer, which aggregates the features learned by the convolutional layers across time, reducing the spatial dimensionality of the output. The output of the average pooling layer is then flattened and passed through several fully connected layers, which learn to classify the input data based on the extracted features. The final layer of the model is a softmax layer for the classification problems and a linear unit for the PVT regression task.

### 3.1.2. LSTM AND BI-LSTM

LSTM and Bi-LSTM networks are alternative architectures to the 1D-CNN to model time-series data. In order to allow the models to capture long-term dependencies in the input signals, we concatenate multiple layers of LSTM or Bi-LSTM units with fully connected layers (see Figure 1(c) and 1(d)).

The first layers of our model are LSTM or Bi-LSTM layers, respectively, which are designed to extract local patterns in the input sequence. After these signal-processing layers, we apply several fully connected layers with dropout regularization. The final layer of the model is a softmax layer for the classification problems and a linear unit for the PVT regression.

## 3.2. Reference Method

As a baseline method that represents the state of the art, we implement all eye-lid movement and gaze-velocity features that we find in the published literature about drowsiness detection (Schleicher et al., 2008; Wilkinson et al., 2013). Table 1 shows a list of base features. The complete set of features is composed of the absolute values for count features, and mean, median, standard deviation, skewness, and kurtosis over all blinks in the input window of all other base features. We train a random forest (RF) classifiers (Breiman, 2001) on these features using the scikit-learn library (Pedregosa et al., 2011).

## 4. Data Collection

This section reports on our data collection. The data set and code are available online[1] and will be published upon acceptance. We record a data set of binocular eye movements and eye-closure features of 47 participants. Participants have been informed about the purpose of the research and the procedure of data collection and have given their informed consent. The study has been approved by the responsible ethics committee. Participants are aged 18 through 48 (mean of 24 years); each participant is recorded in three experimental sessions with a time lag of at least one week in between two sessions. While the participants are instructed to appear well-rested to two of the sessions (*baseline* sessions), one of the sessions takes place under sleep deprivation (*sleep-deprived* session). The order of the experimental conditions is counter-balanced across participants.

---

1. https://osf.io/hmyc4/

Table 1: Engineered features: absolute values for count features and mean, median, standard deviation, skewness, and kurtosis over all blinks in the input window of all other base features.

| | Feature | Source |
|---|---|---|
| 1 | Time steps with eye state "open" (count) | Asaphus Vision |
| 2 | Time steps with eye state "closed" (count) | Asaphus Vision |
| 3 | Time steps with eye state "partially open" (count) | Asaphus Vision |
| 4 | Time steps with eye state "not visible" (count) | Asaphus Vision |
| 5 | Time steps with eye state "downcast" (closed or looking downward, count) | Asaphus Vision |
| 6 | Time steps with eye state "not available" (count) | Asaphus Vision |
| 7 | Number of blinks (count) | - |
| 8 | Blink duration from start to maximum reopening velocity | Schleicher et al. (2008) |
| 9 | Blink duration normalized by mean duration | Schleicher et al. (2008) |
| 10 | Blink duration from maximum closing to maximum opening velocity | Wilkinson et al. (2013) |
| 11 | Blink duration from onset of closing to full reopening | Wilkinson et al. (2013) |
| 12 | Time interval between two adjacent blinks | Schleicher et al. (2008) |
| 13 | Lid-closure amplitude | Schleicher et al. (2008) |
| 14 | Lid-closure amplitude normalized by mean amplitude | Schleicher et al. (2008) |
| 15 | Maximum closure velocity during blink | Schleicher et al. (2008) |
| 16 | Maximum closure velocity during blink normalized by expected velocity | Schleicher et al. (2008) |
| 17 | Mean closure velocity during blink normalized by expected velocity | Schleicher et al. (2008) |
| 18 | Delay between full closure and onset of reopening | Schleicher et al. (2008) |
| 19 | Percentage of time with eyes closed | Wilkinson et al. (2013) |
| 20 | Ratio of the max. amplitude to max. velocity of eyelid movement for the reopening phase | Wilkinson et al. (2013) |
| 21 | Ratio of the max. amplitude to max. velocity of eyelid movement for the closing phase | Wilkinson et al. (2013) |
| 22 | Percentage of time the eyes are fully closed for more than 10 ms | Wilkinson et al. (2013) |
| 23 | Saccade duration | Schleicher et al. (2008) |
| 24 | Saccade duration normalized by mean duration | Schleicher et al. (2008) |
| 25 | Time interval between two adjacent saccades | Schleicher et al. (2008) |
| 26 | Saccade amplitude | Schleicher et al. (2008) |
| 27 | Saccade amplitude normalized by mean amplitude | Schleicher et al. (2008) |
| 28 | Max velocity during saccade | Schleicher et al. (2008) |
| 29 | Max velocity during saccade normalized by expected velocity | Schleicher et al. (2008) |
| 30 | Mean velocity during saccade normalized by expected velocity | Schleicher et al. (2008) |

1. For the *sleep-deprived* session, participants are advised to refrain from sleeping within 24 hours before the experimental session starts, though we do not monitor participants during that time to verify compliance.

2. For each of two *baseline* sessions, participants are asked to appear well rested.

During each of the sessions, participants execute three times the Psychomotor Vigilance Task (PVT) (Dinges and Powell, 1985) (PC-based reimplementation) of 10 minutes,

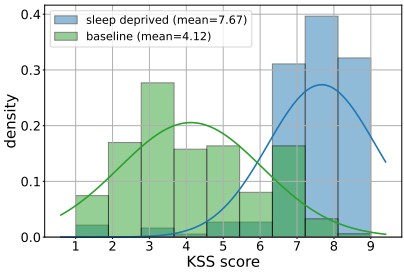
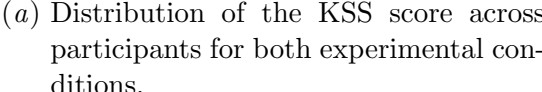
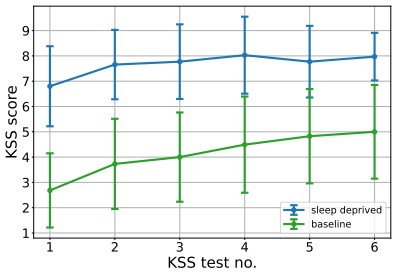

($a$) Distribution of the KSS score across participants for both experimental conditions.

($b$) Development of the mean KSS score across participants in a session over time. Error bars show the standard deviation.

Figure 2: Drowsiness self-assessment with the Karolinska sleepiness scale (KSS).

interrupted by two time intervals (with mean durations of $35 \pm 9$ and $30 \pm 6$ minutes, respectively) in which they perform cognitive and visual tasks for other experiments. We chose this task because it requires sustained attention but no specific skills. Before and after each PVT block, participants report their perceived level of sleepiness on the Karolinska sleepiness scale (KSS), resulting in six KSS scores per session. We linearly interpolate the reported score in order to obtain a sleepiness measure for each point in time.

### 4.1. Technical Setup

We record participants' binocular eye gaze with an Eyelink Portable Duo eye tracker (SR Research) at a sampling frequency of 2000 Hz and a vendor-reported spatial precision of 0.01°. Additionally, we record participants faces with a video-camera, with a sampling frequency of 30 fps and an image resolution of 344×408 px. The camera records in the infrared spectrum and is sensitive to the infrared illumination of the eye tracker (880 nm). During the experiment, participants sit at a height-adjustable table in front of a computer monitor (38×30 cm, 1280×1024 px) with their heads stabilized by a chin and forehead rest.

### 4.2. Descriptive statistics of the recorded data

Figure 2($a$) shows the histogram of reported KSS scores per session type, and Figure 2($b$) over time during each session; in summary, the data cover all levels of drowsiness. For the baseline sessions, the mean KSS score increases from 3 to 5 due to the repetitive nature of the task. In the sleep-deprived session, the mean KSS score increases from 7 to 8.

Figure 3(a) shows the number of microsleep episodes for each reported KSS level as box plot. For any KSS score, zero microsleep events is the mode of the distribution and any data points with microsleep events are outliers. While a correlation between KSS levels and microsleep events is apparent, there is also a large overlap of the distributions of sleep events per minute at different KSS levels, especially for low KSS values.

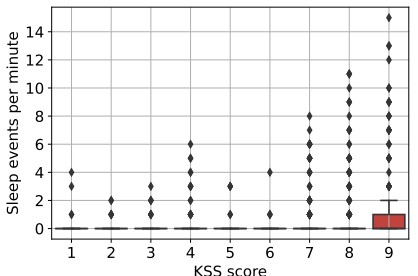
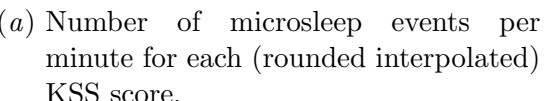

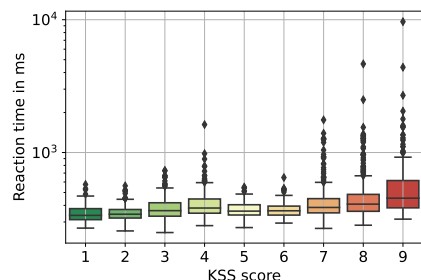

($a$) Number of microsleep events per minute for each (rounded interpolated) KSS score.

($b$) PVT reaction time in ms for each (rounded interpolated) KSS score.

Figure 3: Microsleep and reaction time for each KSS level. Number of microsleep events per minute for each (rounded interpolated) KSS score. In Figures 3($a$) and 3($b$), boxes display the median value and interquartile range; whiskers extend up to the most extreme data point within 1.5 IQR of the quartiles.

Figure 3(b) shows the distribution of PVT reaction times per KSS level; and again, the correlation is apparent. The overlap between distributions for different KSS scores underlines the large inter-person variability. At the KSS level of 9, quite a few still have a lower reaction time than other participants in their fully alert state.

## 5. Experimental Results

This section reports on the evaluation protocol and the experimental results.

### 5.1. Evaluation Protocol

All models are evaluated with a nested five fold cross-validation protocol that is stratified across persons, so that no person appears both in the training and test data at the same time. We tune the hyper-parameters using the training part of the first fold using grid search and use the best found configuration for all remaining folds (see Table 2 for the list of used hyper-parameters and the best found parameters).

### 5.2. Predicting Microsleep is Easier than Predicting KSS or Reaction Time

Table 3 shows that microsleep episodes in the near future can be predicted with an AUC of around 0.95 (0.87 for hard cases in which no prior microsleep events occur in the observation window), whereas self-assessed fatigue is only detected with an AUC of around 0.7. A comparison of the ROC curves in Figure 4($a$) for KSS-sleepiness and Figure 5($a$) and 5($b$) confirms the conclusion that predicting upcoming microsleep events is easier than prediction the KSS self-assessment. The confusion matrix in Figure 4($b$) shows that false-positive and false-negative KSS predictions are more likely to have borderline true scores, but confusions occur across the entire KSS scale.

Table 2: Hyper-parameter grid for the models under investigation and best found values.

| | Hyper-parameter | Search space | Best values for setting | | |
| | | | Microsleep | KSS | Reaction time |
|---|---|---|---|---|---|
| **RF** | Num. of estimators $T$ | {50, 100, 1000} | 100 | 100 | 1000 |
| | Num. of features | {Auto, sqrt, log2} | Auto | sqrt | Auto |
| | Maximum depth of a tree | {2, 4, 6, 8, None} | 8 | None | None |
| | Splitting criterion | {Gini, Entropy} | Gini | Gini | Gini |
| **CNN** | Num. of conv layers $N_C$ | {1, 2, 3} | 1 | 1 | 1 |
| | Kernel size | {16, 32, 64} | [64] | [64] | [64] |
| | Num. of filters | {64, 128} | [128] | [128] | [128] |
| | Stride | {1, 2, 4} | 1 | 2 | 1 |
| | Num. of dense layers $M_C$ | {1, 2} | 2 | 1 | 2 |
| | Num. of hidden dense units | {16, 32, 64} | [64, 32] | [32] | [64, 32] |
| **LSTM** | Num. of LSTM layers $N_{.}$ | {1, 2, 3} | 2 | 2 | 2 |
| | Num. of LSTM units | {16, 32, 64} | [64, 64] | [64, 64] | [64, 64] |
| | Num. of dense layers $M_L$ | {1, 2} | 1 | 1 | 1 |
| | Num. of hidden dense units | {16, 32, 64} | [32] | [32] | [32] |
| **Bi-LSTM** | Num. of Bi-LSTM layers $N_B$ | {1, 2, 3} | 2 | 2 | 2 |
| | Num. of Bi-LSTM units | {16, 32, 64} | [32, 32] | [32, 32] | [32, 32] |
| | Num. of dense layers $M_B$ | {1, 2} | 1 | 1 | 1 |
| | Num. of hidden dense units | {16, 32, 64} | [32] | [32] | [32] |

Table 3: AUC $\pm$ standard error for prediction of the binary KSS label and impending microsleep events. A star indicates models better than the random forest baseline.

| | KSS $\geq 7$ | Microsleep | |
| | | All cases | Hard cases |
|---|---|---|---|
| Random forest | 0.6 ±0.01 | 0.93 ±0.02 | 0.8 ±0.01 |
| CNN | **0.7 ± 0.01**[*] | 0.94 ±0.01 | **0.87 ±0.02**[*] |
| LSTM | 0.67 ±0.04 | **0.95 ± 0.01** | 0.85 ±0.02 |
| Bi-LSTM | 0.66 ±0.04 | 0.93 ±0.01 | 0.82 ±0.03 |

Table 4: Results for predicting the reaction time. A star indicates models better than mean baseline. Mean RMSE $\pm$ standard error and mean $R^2 \pm$ standard error are shown.

| Method | RMSE | $R^2$ |
|---|---|---|
| Mean baseline | 0.27±0.04 | -0.01±0.0 |
| Random forest | 0.26±0.04 | 0.08±0.09 |
| CNN | 0.28±0.02 | -0.83±0.93 |
| LSTM | 0.26±0.04 | 0.06±0.02 |
| Bi-LSTM | **0.25±0.04** | **0.15±0.03** |

Table 4 shows RMSE and $R^2$ metrics for prediction of the PVT reaction time. While AUC, RMSE, and $R^2$ cannot directly be compared, the values show that only 15% of the variance in reaction time can be explained by the KSS level, whereas microsleep episodes in the next 10 seconds can be predicted with an AUC of 0.95. Our interpretation of these

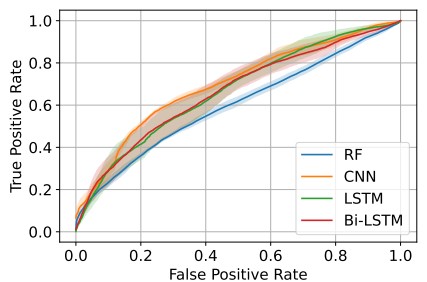
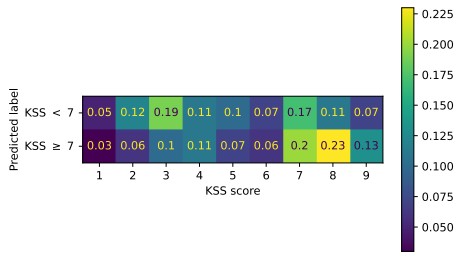

(a) Prediction of binarized KSS score. Shaded bands show the standard error.

(b) Confusion matrix for CNN model for predicting binarized KSS levels and ground truth KSS levels.

Figure 4: Results for the binarized KSS prediction.

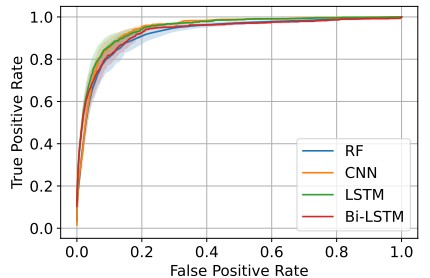
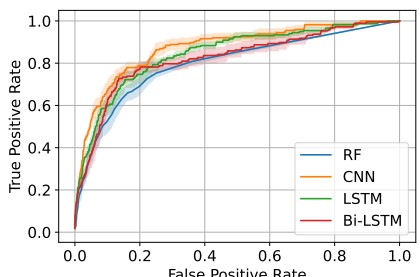

(a) Prediction of microsleep events for all cases.

(b) Prediction of microsleep events for hard cases.

Figure 5: AUC curves for prediction of impending microsleep events. Shaded bands show the standard error.

findings is that impending microsleeps are much more predictable from eye-closure and eye-gaze signals than the KSS self-assessment or the PVT reaction time.

## 5.3. Training on Impeding Microsleep is Better than on KSS Levels

In the next experiment, we apply the models that have been trained to predict impending microsleep and reaction time, respectively, as decision functions for the task of predicting the binarized KSS level. Surprisingly, Table 5 and Figure 6 show that the model that has been trained to predict microsleep seems to be better at predicting the KSS level than the model that has been trained on KSS self-assessments. The model that has been trained to predict reaction time, on the other hand shows a poorer performance at predicting KSS levels than the model trained on KSS levels. However, none of the differences are statistically significant.

Table 5: Cross task evaluation. AUC $\pm$ standard error for predicting the binarized KSS.

|  | Training task | | |
|  | same task | Cross-task | |
|  | KSS $\geq 7$ | Microsleep | Reaction time |
|---|---|---|---|
| Random forest | 0.6 $\pm$0.01 | **0.65 $\pm$0.03** | 0.6 $\pm$0.02 |
| CNN | 0.7 $\pm$ 0.01 | **0.71 $\pm$0.03** | 0.63 $\pm$0.01 |
| LSTM | 0.67 $\pm$0.04 | **0.71 $\pm$0.04** | 0.63 $\pm$0.04 |
| Bi-LSTM | 0.66 $\pm$0.04 | **0.69 $\pm$0.04** | 0.65 $\pm$0.03 |

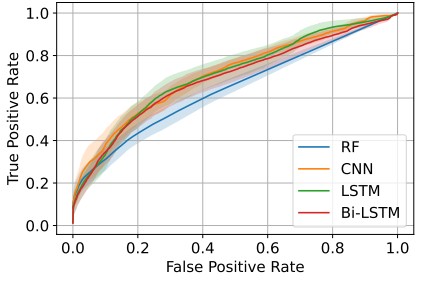

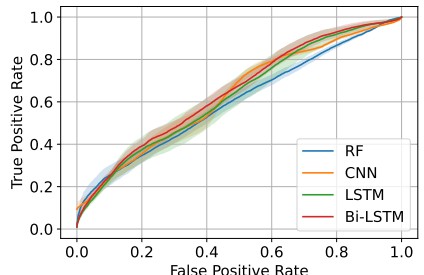

($a$) Model trained to predict impending microsleep events and evaluated on the binarized KSS score prediction.

($b$) Model trained to predict the reaction time and evaluated on the binarized KSS score prediction.

Figure 6: Cross task evaluation. Results for predicting the binarized KSS score. Shaded bands show the standard error.

Our interpretation of these findings is that the presence of microsleep episodes in the near future is a better indicator of sleepiness on the KSS scale than the KSS self-assessment itself. The subjective nature of the self-assessment introduces a high level of noise that renders this signal less useful than the presence or absence of microsleep in the future.

### 5.4. Neural Networks Outperform Engineered Features

For the prediction of microsleep events and KSS levels, Table 3 shows that the neural networks outperform the random forest on engineered features; in two out of three cases, the difference is statistically significant with $p < 0.05$ according to a paired $t$-test. The difference between the alternative network architectures are not significant, but in total the CNN gives the best overall performance picture.

For prediction of the PVT reaction time, the performance of all models is roughly equally poor.

### 5.5. All Signals under Investigation are Useful

Figure 7 and Table 6 show that removing any input channel results in a lower AUC value; removing either the eye-lid channels or the eye-gaze channels results in the lowest AUC values. The deterioration is not statistically significant. Figure 8 shows the engineered fea-

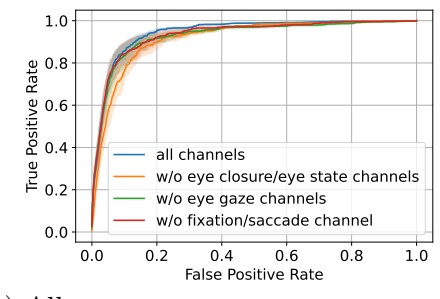

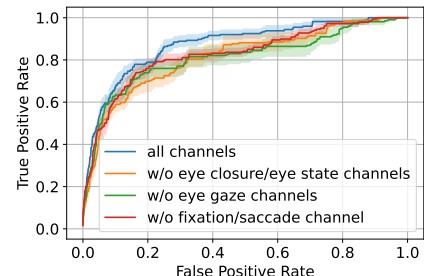

($a$) All cases.

($b$) Hard cases only.

Figure 7: Ablation study. ROC curves for prediction of microsleep events.

Table 6: Ablation study. AUC ± standard error for predicting microsleep events using the CNN model architecture using a subset of input channels.

|  | All | Hard cases |
| --- | --- | --- |
| All channels | 0.94 ±0.01 | 0.87 ±0.02 |
| W/O eye closure and eye state channels | 0.92 ±0.01 | 0.81 ±0.03 |
| W/O eye gaze channels | 0.93 ±0.01 | 0.81 ±0.03 |
| W/O fixation and saccade channel | 0.93 ±0.02 | 0.83 ±0.01 |

tures with highest SHAP value in the random-forest classifier. The figure confirms earlier findings (Schleicher et al., 2008; Wilkinson et al., 2013): the strongest indicators of drowsiness are variability in eye-lid velocity, the percentage of time in which the eyes are closed, variability in blink duration, and delayed reopening during blinks.

## 6. Discussion

The problem setting of drowsiness detection is motivated by efforts to improve the safety of the operation of vehicles and other hazardous machinery. The European New Car Assessment Program (EuroNCAP) has included driver fatigue and incapacitation detection in the catalog of safety functions that affect the safety rating of new vehicles (EuroNCAP, 2021, 2017). The EU General Safety Regulation 2019/2144 makes it mandatory to introduce a range of new safety measures that also include incapacitation detection, following a fixed timetable of stages A-D, scheduled between 2022 and 2029. The current generation of driver cameras that are entering the automotive market are capable of extracting the driver's eye closure and eye gaze in order to detect distraction and drowsiness (Halin et al., 2021). Research on drowsiness detection based on eye-closure and eye-gaze signals therefore has an immediate practical application and potential for societal benefit.

Previous work that applies machine learning to drowsiness detection can be divided with respect to the input signals they use into physiological and vehicle-based approaches. A large body of research to which an overview is given by Stancin et al. (2021) uses EEG signals as predictors for drowsiness. EEG input signals to predictive models are far removed

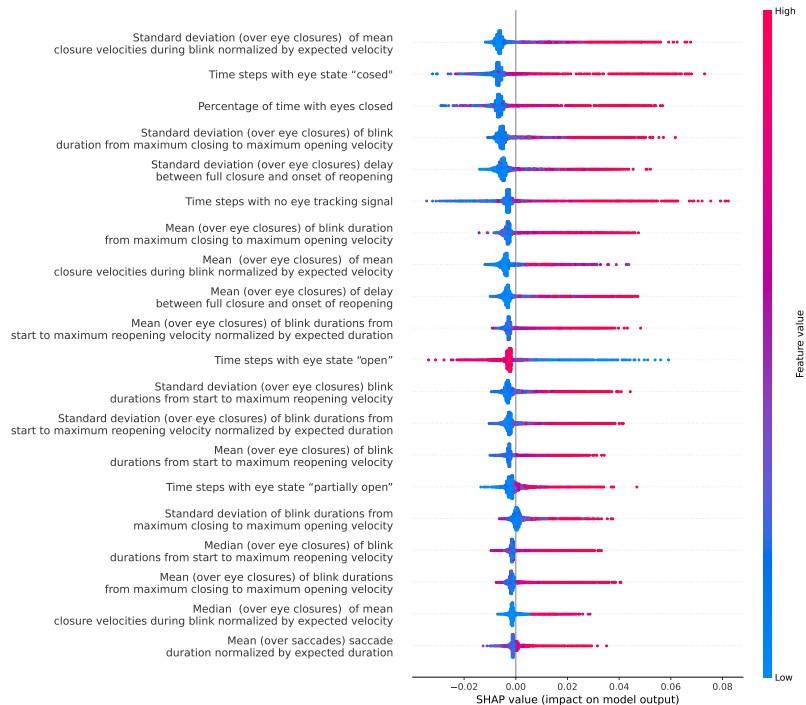

Figure 8: Feature importance for the top 20 features (SHAP values).

from practical applicability since electrodes have to be attached to the head. The same disadvantage applies to electrooculogram (EOG) although it has been found to be more robust against noise (Zhang et al., 2015) compared to EEG.

By contrast, image-based methods have the advantage of being unobtrusive to users. A wide range of image features have been studied, including extract head, facial and eye-lid movements (Schleicher et al., 2008; Wilkinson et al., 2013), raw images of the face (Phan et al., 2021) and eye crops (Quddus et al., 2021). To the best of our knowledge, there is only one other study that uses eye-tracking features as input(Zandi et al., 2019); however, details regarding model and implementation are undisclosed.

Indirect approaches to driver monitoring based on steering wheel interaction (Arefnezhad et al., 2019; Zhenhai et al., 2017) and lane deviation (Friedrichs and Yang, 2010) have been studied extensively and are widely deployed in the market in attention assist systems. This type of indirect monitoring will become insufficient under the EU General Safety Regulation 2019/2144, and will not meet the test criteria of the European New Car Assessment Program (EuroNCAP) from 2024.

A comprehensive comparison of (combinations of) the possible input modalities for drowsiness detection is not available; any such investigation would be hampered by the lack of publicly available data and reference implementations. Nevertheless, it seems plausible that combining modalities such as vehicle interactions, eye closure, eye gaze, head and facial movements, and body posture may add to the robustness of detection systems across all users and their individual characteristics and conditions.

Although EEG is considered the gold standard for the collection of ground-truth labels for sleep stages 2 and higher, it does not allow for an unambiguous detection of drowsiness, and has furthermore the drawbacks of being susceptible to noise caused by muscle movements. While KSS self-assessment is widely regarded to be the gold-standard proxy of ground-truth drowsiness, our findings underscore the subjective nature of this self-assessment that limits the degree of accuracy with which it can be predicted. Analogously, reaction time varies widely across individuals with some persons reacting faster on the brink of sleep than others in their fully alert state. By studying the prediction of impending microsleep, this paper introduces a new proxy of drowsiness ground truth that is objectively hazardous. We interpret the fact that it can be predicted more accurately as indicating that it is a less noisy proxy of actual drowsiness.

With 47 participants, the *Potsdam Binge / PVT data set* data set is not very large by machine-learning standards; it appears likely that a model trained on hundreds or thousands of participants would be considerably more accurate.

## 7. Conclusions

Since drowsiness is an internal state of the mind, the ground truth cannot be observed directly. Based on our experimental findings, we conclude that upcoming microsleep episodes in the near future are a better, less noisy proxy of the ground truth than a self-assessment on the Karolinska sleepiness scale (KSS). Not only can approaching microsleep events be predicted with high accuracy, but a model that has been trained to predict microsleep events is as accurate or even more accurate at predicting a high KSS score than a model that was trained on KSS self-assessments.

We can furthermore conclude that neural network architectures that process the raw eye-state, eye-closure, gaze-velocity, and saccade indicator signals outperform a random forest that processes a comprehensive set of engineered features derived from these signals. The difference in performance between CNN, LSTM, and Bi-LSTM architectures are to small to support any conclusion. Removing any set of features results in a slightly but insignificantly lower performance. The SHAP values that we observe for the engineered features are consistent with earlier findings. However, the neural-networks perform significantly better than the random forest on engineered features, and we therefore conclude that the signal-processing layers have learned to extract additional signals from the raw input that provide evidence of drowsiness.

A large share of research and development on drowsiness detection takes place behind closed doors in the automotive industry and remains unpublished, which impedes the progress of the field as a whole. In order to improve the accessibility of this highly relevant topic to the research community, we share a data set of participants in baseline and sleep-deprived states, a reference implementation of published engineered features, and our implementations of the neural networks.

### Acknowledgments

This work was partially funded by the German Federal Ministry of Education and Research under grant 01|S20043.

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
