# OpenReview forum: "Detection of Drowsiness and Impending Microsleep from Eye Movements"
_NeurIPS.cc/2023/Workshop/Gaze_Meets_ML — Gaze Meets ML 2023 Poster_

### Official Review · Reviewer_GQaS · 2023-10-21
**The authors propose machine learning models to detect drowsiness based on eye movement. The contribution of a new dataset can be useful to other researchers.**

**Rating:** 7
**Confidence:** 3

**Review:**

The writing is clear in general, and the goal of detecting drowsiness when driving is important. The contribution of a new data set is good especially with the PVT and KSS data available. The comparisons of multiple machine learning models and using grid search for the hyperparameters are advantageous. On the other hand, some technical details need to be clarified. More specific comments:

1. The eye movements of drivers may be unavailable or noisy, such as when driving at night or wearing sunglasses. Using eye movements may be infeasible under such situations.
2. In Fig. 1(b) and section 3.1.1, the paragraph mentions that "the output of the average pooling layer is then flattened ...", but the order of the global average pooling and flatten layers seems reversed. Furthermore, if global average pooling is used, the flatten layer is unnecessary, as the output of global average pooling already has the shape of (batch_size, num_channels). Please clarify.
3. As the data are time-series, comparisons with the Transformer's attention mechanism can be interesting.

---

### Official Review · Reviewer_tEBM · 2023-10-21
**Review of Submission #14**

**Rating:** 5
**Confidence:** 4

**Review:**

SUMMARY: This paper explores the use of deep neural sequence models trained person’s raw eye-gaze and eye-closure signals to
detect drowsiness. They examine the prediction of  three measures of drowsiness ground truth: sleepiness self-assessment (Karolinska Sleepiness Scale or KSS), reaction time, and impending microsleep and compare against established baselines that rely on prediction from engineered features. Their analysis indicates that the risk of a microsleep event in the near future can be predicted more accurately than the sleepiness self-assessment or the reaction time. They also assemble a dataset of 47 patients for this task, made publicly available along with the code.

STRENGTHS:

-The problem of drowsiness detection and the relevance of eye gaze measurements is extremely well motivated. Prior literature and the need for data-driven solutions is made very clear in the presentation.
-The open source contributions of the dataset and code could be very valuable to the research community
- Extensive ablation studies have been performed along with post-hoc SHAPLEY investigation to pin-point whether important features concur with the literature.

WEAKNESSES:

- It is not clear why the authors chose a classification task based on the KSS distribution (based on a threshold of 7) rather than a regression or ordinal regression setup. More discussion would be welcome.

- "All models are evaluated with a nested five fold cross-validation protocol that is stratified across persons, so that no person appears both in the training and test data at the same time. We tune the hyper-parameters using the training part of the first fold using grid
search and use the best found configuration for all remaining folds (see Table 2 for the list of used hyper-parameters and the best found parameters)."

Does this mean that only one fold was used for hyper parameter selection? Would this not lead to data leakage for other folds, since the model hyper parameters may have been informed by subjects in the test set of that fold? Please clarify. In a true nested setup, a separate hyper parameter tuning would be performed in each fold.

- Given that there may be a class imbalance for the classification task (since the input KSS scores before thresholding are distributed as Gaussians), why do the authors chose to report only the AU-ROC, as opposed to a more robust measure such as AU-PRC? A suggestion would be to include the full table of results on this task, such as sensitivity, specificity, F1-Score to better indicate how the models are scoring relative to each other on classification tasks.

---

### Official Review · Reviewer_dyvF · 2023-10-24
**Valuable dataset and interesting analysis**

**Rating:** 8
**Confidence:** 4

**Review:**

The authors release a dataset and benchmark results and analysis along with accompanying architectures to tackle the task of drowsiness detection. The authors discuss the benefits of using impending microsleep as an early measure of drowsiness and verify their findings. The authors use a variety of literature-backed handcrafted signals as features to measure performance on the task of drowsiness prediction.
The main novelty and contributions of the work are the released dataset and the accompanying analysis and findings. While the features and proposed models demonstrate reasonable performance, the associated novelty in them is limited.
The authors discuss a few statistics of the collected dataset and discuss the technical details of data collection. Additional discussions around distribution shifts depending on the context of data collection e.g. behavior would be different if the data was collected when subjects are driving.
Overall, a valuable contribution for the community given the utility of the released dataset and associated benchmark models.

---

### Meta-Review · Area_Chair_oUYd · 2023-10-26

**Recommendation:** Accept (Poster)
**Confidence:** 4

**Metareview:**

This paper proposes a study that explores detecting drowsiness through eye movements using deep neural sequence models. It outperforms traditional methods and shows that predicting impending microsleep is more accurate than self-assessment or reaction time, making eye-closure and gaze information valuable in drowsiness detection. Additionally, the study shares eye-gaze data to advance research on this topic.

The results are promising, and the dataset is released for the benefit of the community. The authors should address reviewers' concerns regarding the evaluation procedures and provide more details.

---

### Decision · Program_Chairs · 2023-10-26

Accept (Poster)